# Outdoor physical activity, residential green spaces and the risk of dementia in the UK Biobank cohort

Benjamin Kröger [1,2,3], Hui-Xin Wang[4], Örjan Ekblom[1,5], Jing Wu[6], Hugo Westerlund[4], Mika Kivimäki[7,8] & Rui Wang [1,2,9] ✉

## Abstract

**Background** How the environment in which physical activity takes place influences brain health remains insufficiently studied. We aimed to investigate the association between outdoor physical activity and dementia in older adults, and to examine whether residential green space (GS) modifies this association.

**Methods** This prospective longitudinal study used UK Biobank data from 187,724 adults aged 60—73 years at baseline (2006—2010) and 36,854 with brain Magnetic Resonance Imaging (2014—2021). Outdoor activity (via Metabolic Equivalent of Task) and GS (within 300 m of homes) were assessed, with dementia incidence tracked through electronic records until December 2022. Neuroimaging markers included hippocampal and total gray-matter volumes, and white-matter hyperintensity volume. Cox proportional hazards and linear regression models were applied.

**Results** During a mean follow-up of 13.01 years, 7218 participants (47.1% female) developed dementia. After adjusting for covariates, higher outdoor activity is associated with a lower risk of all-cause dementia (HR$_{top vs bottom quartile}$ = 0.84, 95% Confidence Interval [CI] 0.78—0.90). This association is most pronounced for vascular dementia (HR:0.72, 95% CI 0.63—0.83) and is stronger among those living in areas with high, compared to low, residential GS (0.73, 95%CI 0.63—0.85 vs 0.86, 95%CI 0.79—0.93, p$_{interaction}$ = 0.04). Higher outdoor activity is also associated with higher hippocampal and total gray-matter volumes and fewer white-matter hyperintensities. A combined effect of outdoor activity and GS on hippocampal volume is observed.

**Conclusions** Outdoor activity is linked to lower dementia risk, particularly in those living in more accessible GS. These findings underscore the importance of urban planning that prioritizes accessible GS to promote brain health in the aging population.

## Plain language summary

Spending time outdoors and being active, especially in natural environments, is known to benefit mental health. Some studies suggest that access to green spaces may also help protect against dementia, but long-term effects and possible brain-related explanations are not well understood. In this study, we followed over 180,000 older adults for 13 years and found that those engaging in more outdoor physical activity had a lower risk of developing dementia. This protective effect was stronger for people living in neighbourhoods with more accessible green spaces. Outdoor activity was also linked to better brain structure. These findings suggest that increasing outdoor activities for older adults—especially in green areas—could help support brain health as they age.

Approximately 50 million individuals worldwide are suffering from dementia, and this number is expected to reach 150 million by 2050[1]. As dementia is accompanied by severe disability, dependency, and a high burden on the individual, relatives, economy, and society, it is essential to identify preventive approaches against dementia[1]. Physical activity has been shown to benefit cognition and prevent or delay the onset of dementia[2,3]. However, it remains unclear how the settings or environment in which the exercises occur may influence dementia prevention.

Green exercise refers to physical activity that occurs in natural settings, allowing direct exposure to green environments[4]. It includes outdoor

[1]Department of Physical Activity and Health, The Swedish School of Sport and Health Science, Stockholm, Sweden. [2]Division of Clinical Geriatrics, Department of Neurobiology, Care Sciences and Society, Karolinska Institutet, Stockholm, Sweden. [3]National Graduate School on Ageing and Health (SWEAH), Lund, Sweden. [4]Stress Research Institute, Department of Psychology, Stockholm University, Stockholm, Sweden. [5]Department of Neurobiology, Care Sciences and Society, Division of Nursing, Karolinska Insitutet, Huddinge, Sweden. [6]Institute of Environmental Medicine, Karolinska Institutet, Stockholm, Sweden. [7]Brain Sciences, University College London, London, UK. [8]Clinicum, University of Helsinki, Helsinki, Finland. [9]Wisconsin Alzheimer's Disease Research Center, University of Wisconsin School of Medicine and Public Health, Madison, WI, USA. ✉e-mail: rui.wang@gih.se

activities, such as walking, cycling, horse riding, fishing, boating, and gardening, all of which involve exposure to green or natural environments[4,5]. Acute effects of outdoor activities on brain health have been demonstrated in experimental studies involving healthy adults. For example, one study showed that a 15 min walk outdoors was associated with improved cognitive performance, whereas a 15 min indoor walk was not[6]. Another study, using whole-brain analysis of neuroimaging data, found that certain time spent outdoors was positively associated with greater gray-matter volume in the right dorsolateral prefrontal cortex, independent of physical activity levels[7]. A recent study compared neuroimaging markers before and after walks in either a natural or urban environment and found that hippocampal subfield volume increased after a one-hour walk in nature, but showed no change after a walk in the urban environment[8]. These findings suggest that outdoor activities, particularly those in natural environments, may have a more positive effect on the brain than indoor activities[6,7,9]. Systematic reviews, however, highlight the low quality of evidence regarding the long-term health benefits of outdoor activities in nature and call for more robust research designs[5,9]. In particular, the association between outdoor activities and long-term dementia risk remains underexplored. Moreover, to better understand the underlying mechanisms linking outdoor activities to dementia, more evidence is needed on how these activities influence age-related brain health markers, such as neuroimaging markers of neurodegeneration and cerebrovascular diseases.

In addition, previous large cohort studies have indicated that greater residential green space (GS) is associated with a reduced risk of cognitive decline and dementia[10–14]. These studies suggest that the association may be explained by reduced chronic disease burden and slower vascular aging, including arterial stiffening, reduced cerebral blood flow, and endothelial dysfunction[10–14]. It is also plausible that surrounding residential environments may encourage outdoor activity by motivating and facilitating physical activity, thereby further reducing vascular aging and the long-term risk of dementia[15–18]. However, evidence, particularly from large cohort studies, on whether residential GS modifies the association between outdoor activity and dementia risk or age-related brain health in older adults remains limited.

Utilizing data from the UK Biobank study, we sought to (1) quantify the association between outdoor activity and 13-year risk of incident dementia at older ages (≥ 60 years at baseline); (2) examine whether residential GS modifies this association; and (3) explore potential underlying mechanisms by linking outdoor activity to neuroimaging markers. We observe that greater engagement in outdoor activity is associated with a reduced risk of overall—and especially vascular—dementia. Outdoor activity is also linked to brain structures, including larger hippocampal and gray-matter volumes and fewer white-matter hyperintensities. The protective association is stronger among individuals living in neighbourhoods with more accessible GS, underscoring the potential of outdoor activity in natural environments to support brain health during aging.

## Methods
### Design
This study is based on the prospective longitudinal cohort—the UK Biobank. At baseline of the UK Biobank (2006—2010), over 500,000 participants aged ≥40 years attended to clinical examination in one of the 22 assessment centres throughout the UK[19]. At the baseline data collection, personal and environmental information was obtained through a touchscreen questionnaire and a structured interview. Physical and functional tests were conducted, and samples of biofluids (blood, urine, saliva) were obtained from study participants. Data collection on neuroimaging markers started in 2014, and approximately 76,000 individuals attended the first neuroimaging visit. Participants' health-related outcomes were tracked by linking them with nationally available datasets, such as primary care, hospitalisation, and cause-specific death registries[19].

Among 502,198 participants at enrolment, we excluded participants with dementia prior to the baseline assessment (n = 258), without valid follow-up data (n = 39,170), or missing value in physical activity data

(n = 43,752), leaving 419,018 individuals for further analysis. Of the 419,018 participants, we involved 231,294 participants who aged ≥60 years to quantify the association between outdoor activities and dementia risk, and to examine whether GS modifies this association. Given that pathological brain changes may occur decades before a dementia diagnosis, we included all participants who underwent MRI scanning to explore potential underlying mechanisms of the observed association. This analysis was based on an analytical sample of participants with available brain neuroimaging data (n = 36,854). A flow chart of sample selection is shown in Fig. 1.

This study follows the Strengthening the Reporting of Observational Studies in Epidemiology (STROBE) reporting guideline.

### Measures
Measurement of outdoor activities was conducted at baseline (2006–2010) and at the neuroimaging visit (2014–2022). We derived the frequency and duration of different activity intensities conducted from the sub-domains assessed in the short form of the International Physical Activity Questionnaire (IPAQ)[20], encompassing leisure, domestic, work-related, and transport activities. The evaluation of outdoor activities was based on the following items: (1) walking for pleasure, (2) participating in light DIY activities (do-it-yourself, e.g., home improvement and gardening), (3) undertaking heavy DIY tasks (e.g., using heavy tools, weeding, lawn mowing, digging, and carpentry), and (4) participating in other forms of exercise (e.g., swimming and cycling). We quantified outdoor activities as weekly metabolic equivalent of task (MET) minutes in total and in different items (see Supplementary Data 1).

Measurement of GS was conducted at baseline in 2006–2010. Residential GS exposure was estimated as the percentage of green areas in 300 m buffer zones, surrounding the home location of the participants[13]. Specifically, using the 2005 Generalised Land Use Database in England, identifiable land features were categorized into groups, including domestic and non-domestic buildings, roads, paths, rail, domestic gardens, green areas, water, other land uses, and unclassified. The percentage of GS was determined by calculating its proportion relative to the total percentage of all land use types within each census output area. We applied the area-weighted mean of GS percentage coverage for each participant by intersecting census output area classifications with their home locations and the 300 m buffer zones around them. The same procedure was used to incorporate residents of Wales and Scotland, utilizing data from the 2007 Land Cover Map[21].

Our follow-up of dementia through linkage to electronic health records was until December 2022. All-cause dementia, as well as its subtypes, is obtained from algorithmically defined cases, the first-occurrence data reported dementia onsets, death register, hospital inpatient record, and primary care data recorded[22]. All-cause dementia, Alzheimer's disease (AD), vascular dementia, and frontotemporal dementia were classified according to the International Classification of Diseases (ICD) codes (see Supplementary Data 1)[22,23].

We applied neuroimaging markers derived from structural sequences between 2014 and 2021, including T1-weighted and T2-weighted images acquired using a Siemens Magnetom Skyra 3 T scanner[24,25]. Two types of neuroimaging markers were applied in our analyses, including neurodegenerative markers (total gray-matter volume and hippocampal volume) and one marker of cerebral small vessel disease—white-matter hyperintensity (WMH). Based on the T1 and T2-FLAIR sequence, total gray-matter volume, hippocampal volumes from left and right hemispheres, and total WMH volume were generated by an image-processing pipeline developed and run on behalf of UK Biobank, which were available as part of the UKB central analysis (https://biobank.ctsu.ox.ac.uk/crystal/crystal/docs/brain_mri.pdf)[24,25]. In our data analyses, we involved the gray-matter volume normalized for head size, and the total hippocampal volume from the left and right hemispheres weighted by total brain tissue volume to examine the association between outdoor activities, GS, and markers of neurodegeneration. For the WMH analyses, we applied the log-transformed WMH volume to ensure normal distribution.

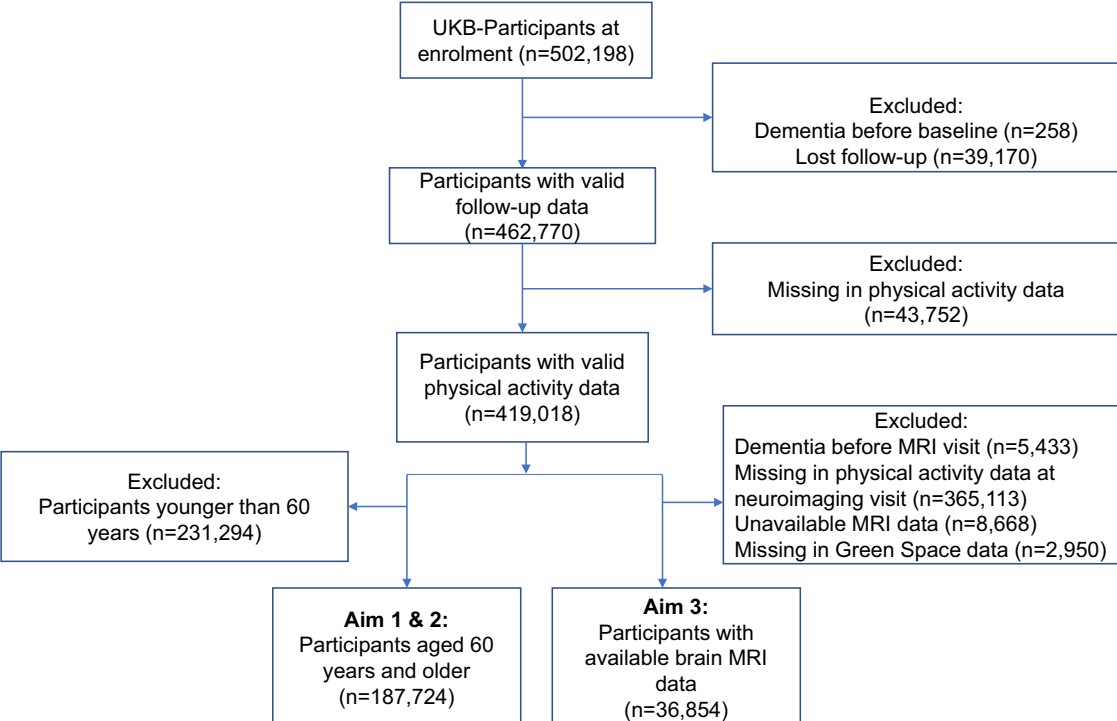

**Fig. 1 | Flow chart of study participants.** *Notes*. This flow chart illustrates how the study samples were derived from the original UK Biobank participants. Aim 1 is to quantify the association between outdoor activity and the risk of incident dementia among participants aged 60 years at baseline. Aim 2 is to examine whether residential green space modifies this association. Aim 3 is to explore potential underlying mechanisms by linking outdoor activity to neuroimaging markers. Abbreviations. MRI Magnetic Resonance Imaging.

Covariates, including sociodemographic, health, and residential characteristics, were assessed at baseline (2006–2010). The baseline age of the participants was determined by subtracting their birth year and month from the assessment date. Biological sex was categorized into female and male. The participants' education level was divided into the categories "with a college or university degree or higher" and "without a college or university degree". Smoking status was categorized as current, former, or never. Body mass index (BMI) was calculated using height and weight and then classified into four groups: underweight ($< 18.5 \, kg/m^2$), normal ($18.5 \leq BMI < 25 \, kg/m^2$), overweight ($25 \leq BMI < 30 \, kg/m^2$), obese ($\geq 30 \, kg/m^2$)[26]. Based on multiple resources (e.g., blood markers in HbA1c, blood pressure, total cholesterol, use of medications, self-reported diagnosis), we additionally considered diabetes mellitus (present/not present), high cholesterol (present/not present), and hypertension (present/not present) to assess cardiovascular health. We used the Charlson Comorbidity Index (CCI), which summarizes 19 medical conditions based on ICD-10 codes from UK Biobank inpatient data, to assess participants' somatic health[26]. Residential length in years was self-reported by participants at baseline. Residential social and economic status was evaluated using the Index of Multiple Deprivation (IMD), which was matched to the baseline assessment according to the national census areas considering the participant's home location. Air pollution estimates for the years 2005–2007 were derived from EU-wide air pollution maps, with a resolution $100 \times 100$ m. Noise estimates for the year 2009 were modelled using a version of the CNOSSOS-EU noise model, and we calculated 24 h noise pollution by averaging the mean daytime, evening and nighttime levels of noise pollution.

**Statistics and reproducibility**
The baseline characteristics of the total sample and subsamples, stratified by incident dementia, were presented with frequencies (n, %) for categorical variables, means (standard deviation [SD]) for normally distributed continuous variables, and medians (interquartile range) for non-normally distributed continuous variables. The differences in characteristics between demented and non-demented participants were compared using chi-square tests for categorical variables or t-test/Wilcoxon test for continuous variables.

Cox proportional hazards regression modelling, using follow-up time as the underlying time variable, was utilized to examine the relationship between outdoor activities and all-cause and cause-specific dementia. Follow-up time was calculated as the time from baseline assessment until the first event of dementia, death, or December 31, 2022, whichever occurred first. The sum of MET minutes of outdoor activities was modelled either as quartiles or as a log-transformed continuous variable to capture potential non-linear and linear associations with dementia outcomes. Two models were involved here. Model 1 was adjusted for individual-level factors (e.g., demographic factors, somatic health), and Model 2 was additionally adjusted for residential factors (e.g., air pollution and noise levels). The BMI variable was not included in the models, as it violated the proportional hazards assumption. The modifying effect of GS on the association between outdoor activities and dementia was investigated by introducing an interaction term between outdoor activities and GS into the models. When a significant interaction was observed, we further conducted stratification analyses by GS to examine whether and to what extent the relationships between outdoor activities and dementia were magnified by more accessible residential GS. In the stratification analyses, three models were employed. Model 1 included adjustments for individual-level factors (sex, age, education, smoking, cardiovascular health, somatic health, and residential length). Model 2 extended the adjustments of Model 1 to incorporate the residential socioeconomic factor (IMD). Model 3 was additionally adjusted for residential-level physical factors (air pollution and noise level). In this study, missing data on confounding variables were addressed by including them as separate dummy categories in the regression models.

The associations between GS and outdoor activities, including specific activity items, were presented as deciles using density correlation plots. We employed ordinal and multinomial logistic regression models to delve

deeper into the association between activities and GS deciles, adjusting for both individual-level and residential-level factors.

To explore the underlying mechanisms linking outdoor activities and GS to the dementia risk reduction, further analyses were performed among those aged ≥40 at the baseline and with available neuroimaging markers. Multivariable linear regression models were applied to estimate the $\beta$-coefficients and 95% confidence intervals (CI) of neuroimaging markers linking to outdoor activities and residential GS.

We performed all analyses in R software (version 4.2.2). The *survival* package was used for Cox regression models. *MASS* and *nnet* packages were used for ordinal logistic regressions and multinominal logistic regressions respectively. The *stats* package was used for the multiple linear regressions.

### Patient and public involvement
Details of the patient and public involvement in the UK Biobank are available online (www.ukbiobank.ac.uk/about-biobank-uk/) and (https://www.ukbiobank.ac.uk/wp-content/uploads/2011/07/Summary-EGF-consultation.pdf?phpMyAdmin=trmKQlYdjjnQIgJ%2CfAzikMhEnx6). No patients were specifically involved in setting the research question or the outcome measures, nor were they involved in developing plans for recruitment, design, or implementation of this study. No patients were asked to advise on interpretation or writing up of results. There are no specific plans to disseminate the results of the research to study participants, but the UK Biobank disseminates key findings from projects on its website. The funders of the study had no role in study design, data collection, data analysis, data interpretation, or the writing of the report.

### Ethical approval
All UK Biobank participants signed informed consent before information collection and the study has full ethical approval from the NHS National Research Ethics Service (16/NW/0274). UK Biobank's creation and operations have received ethics approval from the North West Multi-centre Research Ethics Committee (MREC). This research has been conducted using the UK Biobank Resource under Application Number 86931.

## Results
### Baseline characteristics of the study participants
The average age of the 187,724 participants was 64.2 years (range: 60–73 year), and 52% were female (Table 1). During the follow up period (mean: 13.01 years with SD 2.52 years), 7218 individuals developed dementia. Participants who developed incident dementia differed from those remaining dementia-free in all individual and residential factors measured, except the average noise levels and residential length.

### Association between outdoor activity and risk of incident dementia
The multivariable-adjusted Cox regression model showed that higher MET minutes/week of outdoor activities were associated with a significantly lower risk of incident all-cause dementia (hazard ratio [HR] = 0.93; 95% CI: 0.91—0.95, $p < 0.001$). Similar results were observed for vascular dementia (HR = 0.88, 95% CI: 0.84—0.92, $p < 0.001$) and AD (HR = 0.96, 95% CI: 0.93—1.00, $p = 0.03$), although the association with frontotemporal dementia did not reach statistical significance (HR = 0.90, 95% CI: 0.78—1.04, $p = 0.15$). The findings related to frontotemporal dementia may be due to insufficient statistical power. We divided outdoor activity into quartiles to further explore its potential non-linear association with dementia outcomes. Results indicated that higher quartiles of outdoor activity were associated with greater risk reductions for all-cause dementia, AD, and vascular dementia, with the strongest effects observed in the second quartile and smaller gains in the higher quartiles (Fig. 2).

### Association between outdoor activity and residential GS
Higher accessibility to residential GS was associated with a higher level of general engagement in outdoor activities (Fig. 3). This association was particularly clear for engagement in activities such as walking for pleasure, light DIY, and heavy DIY. Results from the ordinal and multinomial logistic regression analysis suggested that a higher density in GS was associated with increased engagement in outdoor activities.

### Modifying effect of residential GS on the association between outdoor activity and incident dementia
An interaction of residential GS and outdoor activity on all-cause dementia was observed (p value for interaction = 0.045), after adjusting individual- and residential-level covariates in the Cox model. Further stratified analyses were conducted between participants with more accessibility to residential GS (top quartile group) and those with less accessibility to residential GS (remaining quartile groups). Figure 4 presents the graphical results, while Supplementary Table 1 shows the detailed HRs and 95% CIs from the multivariable-adjusted model. Compared to the lowest quartile of total outdoor activities, the highest quartile was significantly associated with a reduced risk of dementia, with a greater magnitude of risk reduction observed in areas with more accessible GS (more accessible GS: $HR_{Q4\ vs.\ Q1} = 0.73$, 95% CI = 0.63–0.85, $p < 0.001$; less accessible GS: $HR_{Q4\ vs.\ Q1} = 0.86$, 95% CI = 0.79–0.93, $p < 0.001$). When examining individual outdoor activities, we observed that the associations of walking and light DIY with dementia risk varied slightly depending on the amount of residential GS. The association between higher quartiles of light DIY and reduced dementia risk appeared more pronounced among individuals living in areas with more accessible GS (more accessible GS: $HR_{Q4\ vs.\ Q1} = 0.77$, 95% CI = 0.63–0.95, $p = 0.02$; less accessible GS: $HR_{Q4\ vs.\ Q1} = 0.93$, 95% CI = 0.83–1.04, $p = 0.21$). Walking was not associated with dementia risk in areas with more accessible GS; however, compared to the lowest quartile, the highest quartile of walking was associated with an increased risk of dementia among those living in areas with less GS ($HR_{Q4\ vs.\ Q1} = 1.15$, 95% CI = 1.05 — 1.26, $p = 0.003$).

### Associations of outdoor activity with neuroimaging markers by residential GS
Among the analytical sample with available neuroimaging data (individuals who were ≥40 years old at baseline), we linked outdoor activities to neuroimaging markers. The characteristics of this sample were displayed in Supplementary Table 2. Results showed that increased engagement in outdoor activities was associated with larger hippocampal volume, larger total gray-matter volume, and fewer WMH, while adjusting for both individual-level factors and residential GS (*Model 1 in* Table 2). When stratifying the analyses based on the level of residential GS, the results indicated that the association between outdoor activities and neurodegenerative markers, especially hippocampal volume, was more pronounced in individuals residing in areas with more GS. However, the association between outdoor activity and WMH did not differ based on residential GS.

### Sensitivity and additional analyses
We considered other residential nature environments regarding their modifying effect on outdoor activities and dementia risk reduction, such as blue space (e.g., a lakeside, river or coast), domestic garden percentage, and natural environment. Similar results were observed for these natural environments (Supplementary Table 3). Additional analyses using residential GS within a 1000 m buffer zone yielded similar results (Supplementary Figs. 1 and 2, Supplementary Table 3 and 4). To address reverse causality, we excluded incident dementia cases recorded during the first 10 years of follow-up[27], and the results remained consistent with those from the main analysis (Supplementary Table 5).

## Discussion
This analysis of the UK Biobank data yielded three main findings: First, a higher level of outdoor activities among older adults was associated with a lower risk of all-cause dementia, especially vascular dementia and AD. Second, residential GS appeared to modify this association, with the reduced risk of dementia associated with outdoor activities being more pronounced in older adults residing in areas with high accessibility to GS. Third,

**Table 1 | Baseline characteristics of study participants by incident dementia**

| Characteristics | Total | Incident dementia | | p-value |
|---|---|---|---|---|
| | (N = 187 724) | Yes (n = 7218) | No (n = 180 506) | |
| Age (years), mean (SD) | 64.2 (2.9) | 65.7 (2.7) | 64.1 (2.8) | <0.001 |
| Sex, n (%) | | | | |
| Female | 97,620 (52.0) | 3398 (47.1) | 94,222 (52.2) | <0.001 |
| Male | 90,104 (48.0) | 3820 (52.9) | 86,284 (47.8) | |
| Education, n (%) | | | | |
| Without a college or university degree | 135,066 (71.9) | 5564 (77.1) | 129,502 (71.7) | <0.001 |
| College or university degree or higher | 50,296 (26.8) | 1513 (21.0) | 48,783 (27.0) | |
| Data missing | 2362 (1.3) | 141 (2.0) | 2221 (1.2) | |
| Smoking Status, n (%) | | | | |
| Current | 14,354 (7.6) | 620 (8.6) | 13,734 (7.6) | <0.001 |
| Never | 93,682 (49.9) | 3333 (46.2) | 90,349 (50.1) | |
| Previous | 78,856 (42.0) | 3217 (44.6) | 75,639 (41.9) | |
| Unknown | 832 (0.4) | 48 (0.7) | 784 (0.4) | |
| BMI categories, n (%) | | | | |
| Underweight (< 18.5) | 778 (0.4) | 43 (0.6) | 735 (0.4) | |
| Normal (18.5 ≤ BMI < 25) | 55,777 (29.7) | 2203 (30.5) | 53,574 (29.7) | 0.025 |
| Overweight (25 ≤ BMI < 30) | 85,837 (45.7) | 3229 (44.7) | 82,608 (45.8) | |
| Obese (≥ 30) | 44,612 (23.8) | 1697 (23.5) | 42,915 (23.8) | |
| Data missing | 720 (0.4) | 46 (0.6) | 674 (0.4) | |
| Diabetes, n (%) | 13,993 (7.5) | 944 (13.1) | 13,049 (7.2) | |
| High Cholesterol, n (%) | 107,987 (57.5) | 4490 (62.2) | 103,497 (57.3) | |
| Hypertension, n (%) | 127,991 (68.2) | 5375 (74.5) | 122,616 (67.9) | |
| Outdoor activities (100 MET-min/week), median (IQR) | 9.0 (3.7–18.9) | 8.2 (3.3–17.7) | 9.0 (3.8–18.9) | <0.001 |
| Outdoor activities categories, n (%) | | | | |
| First quartile (< 365.6) | 46,931 (25.0) | 2000 (27.7) | 44,931 (24.9) | <0.001 |
| Second quartile (365.5 ≤ GE < 900.0) | 46,931 (25.0) | 1831 (25.4) | 45,100 (25.0) | |
| Third quartile (900.0 ≤ GE < 1888.9) | 46,931 (25.0) | 1713 (23.7) | 45,218 (25.1) | |
| Fourth quartile (≥ 1888.9) | 46,931 (25.0) | 1674 (23.2) | 45,257 (25.1) | |
| GS (300 m buffer) in %, mean (SD) | 36.7 (23.7) | 36.8 (23.8) | 35.4 (22.5) | <0.001 |
| GS quartiles 300 m buffer, n (%) | | | | |
| First quartile (< 18.0) | 41,713 (22.2) | 1559 (21.6) | 40,154 (22.2) | <0.001 |
| Second quartile (18.0 ≤ GS < 31.1) | 41,713 (22.2) | 1704 (23.7) | 40,009 (22.2) | |
| Third quartile (31.1 ≤ GS < 50.8) | 41,713 (22.2) | 1736 (24.1) | 39,977 (22.1) | |
| Fourth quartile (≥ 50.8) | 41,713 (22.2) | 1385 (19.2) | 40,327 (22.3) | |
| Data missing | 20,873 (11.1) | 834 (11.6) | 20,039 (11.1) | |
| IMD, mean (SD) | 15.9 (13.0) | 17.7 (14.4) | 15.8 (12.9) | <0.001 |
| Air Pollution (PM$_{10}$), mean (SD) | 21.9 (2.8) | 21.9 (2.7) | 21.9 (2.8) | 0.032 |
| Noise Pollution (LAeq), mean (SD) | 51.1 (4.2) | 51.1 (4.2) | 51.1 (4.2) | 0.724 |
| Residential length (years), median (IQR) | 23.0 (10.0–32.0) | 23.0 (10.0–33.0) | 23.0 (10.0–32.0) | 0.170 |

*SD* Standard Deviation, *BMI* Body Mass Index, *IQR* Interquartile Range, *GS* Green Space, *IMD* Index of Multiple Deprivation, *PM* Particulate Matter, *LAeq* A-weighted equivalent sound level.

supporting these findings, higher engagement in outdoor activities—particularly among individuals living in areas with high residential GS—was associated with better preservation of hippocampal volumes.

Most previous studies focused on outdoor activity in people living with dementia. It has been hypothesized that engagement in activities outside of nursing homes might benefit patients with dementia in diverse ways[16,28]. A systematic review focusing on intervention studies involving outdoor activities, e.g., aquatic exercise, wheelchair cycling, walking, and outdoor gardening, found preliminary evidence of benefits for individuals living with dementia in terms of psychological, physical, and physiological outcomes[28].

However, the limited number of participants in the included studies for the joint analysis (n = 177) may restrict the generalizability of the findings[28]. A narrative review focusing on long-term care facilities for individuals with dementia evaluated studies examining the impact of outdoor natural landscapes and activities, e.g., such as horticulture and gardening, on behavioural and psychological symptoms of dementia[16]. The review summarized that outdoor natural landscape interventions—such as horticultural activities, therapeutic gardens, and green care farms—generally showed positive effects on agitation, apathy, and engagement among individuals with dementia[16]. To our knowledge, the current study is the first of its

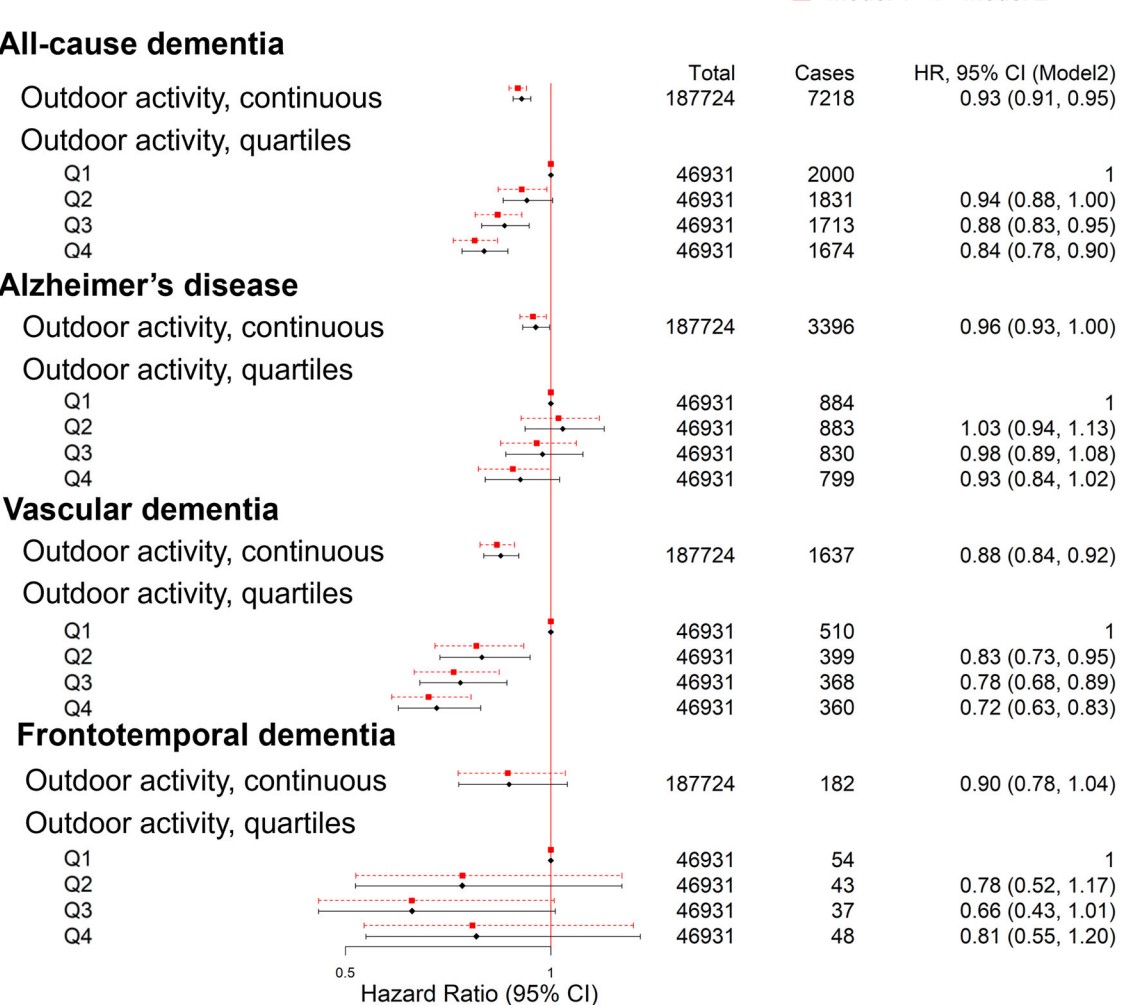

**Fig. 2 | Hazard Ratio (95% confidence interval) of all-cause dementia and cause-specific dementia in relation to outdoor activity.** Model 1 was adjusted for individual-level factors (age, sex, education, smoking, cardiovascular health, somatic health, and residential length). Model 2 was adjusted for individual-level factors, index of multiple deprivations, air pollution, and noise level. The unit value for the continuous variable of outdoor activity was based on per log-transformed 100 MET-min/week. Abbreviations. HR hazard ratio, CI confidence interval.

kind to leverage a large population sample to explore the link between outdoor activities and long-term dementia risk.

Based on twenty-two papers published between 2012 and 2020, a recent review concluded that greater exposure to residential GS correlates with a reduced risk of cognitive impairment and dementia[29]. Furthermore, an apparent protective effect of long-term GS exposure on cognition was observed throughout the lifespan[30]. Using the same dataset from the UK Biobank, one study observed that more accessible residential GS was associated with all-cause dementia, especially vascular dementia in middle-aged and older adults[13]. A further population-based cohort of 1.7—4.3 million adults in Canada showed that increased exposure to urban GS was associated with reduced incidence of both dementia and stroke[11]. The current study extends these findings by showing that among older adults aged 60 years and older, accessible neighborhood GS, as well as other nature components (e.g., rivers, lakes, oceans, farmlands, meadows), may magnify the impact of outdoor activities on dementia risk reduction. In addition, we investigated the potential for greater accessibility to GS to promote greater engagement to outdoor activities by conducting an association analysis between residential GS and outdoor activities. The results suggested a tendency to confirm such relationships, aligning with previous findings[31].

The World Health Organisation's Age-Friendly Cities framework, outlined in the Global Age-friendly Cities Guide[32], proposes eight interconnected domains to address barriers to the well-being and participation of older adults. The domain of outdoor spaces and buildings has gained increased attention from public health experts and stakeholders. A city with well-maintained recreational areas offers an ideal setting for seniors to age in place. In the UK, Middlesbrough Council has joined this Global Network for Age-friendly Cities and Communities, aiming to create a town where older people can live healthy and active lives.

Findings from the current study indicate that the availability of residential natural and GS plays a key role in promoting outdoor activities among older individuals, thereby reducing the risk of dementia over a mean follow-up of 13 years and benefiting brain health. Our findings highlight the importance of urban planning that prioritizes accessible, well-designed GS to promote healthy behaviours and brain health in the aging population.

Our study indicated a link between outdoor activities, residential GS, and dementia. Future research is needed to apply objective measurements of outdoor activity, particularly wearable tools, to better understand its connections to dementia and brain health. This approach would more effectively capture the combined effect of exposure to nature and outdoor activities, providing robust evidence. Research on the pathways or mechanisms underlying the potential beneficial effects of outdoor activities and exposure to nature on dementia and subtypes of dementia is scarce. Previous research has developed the conceptual model suggesting that physical activities in natural environments may enhance cognition through sensory stimulation, cortisol secretion, mindfulness, neural plasticity, and

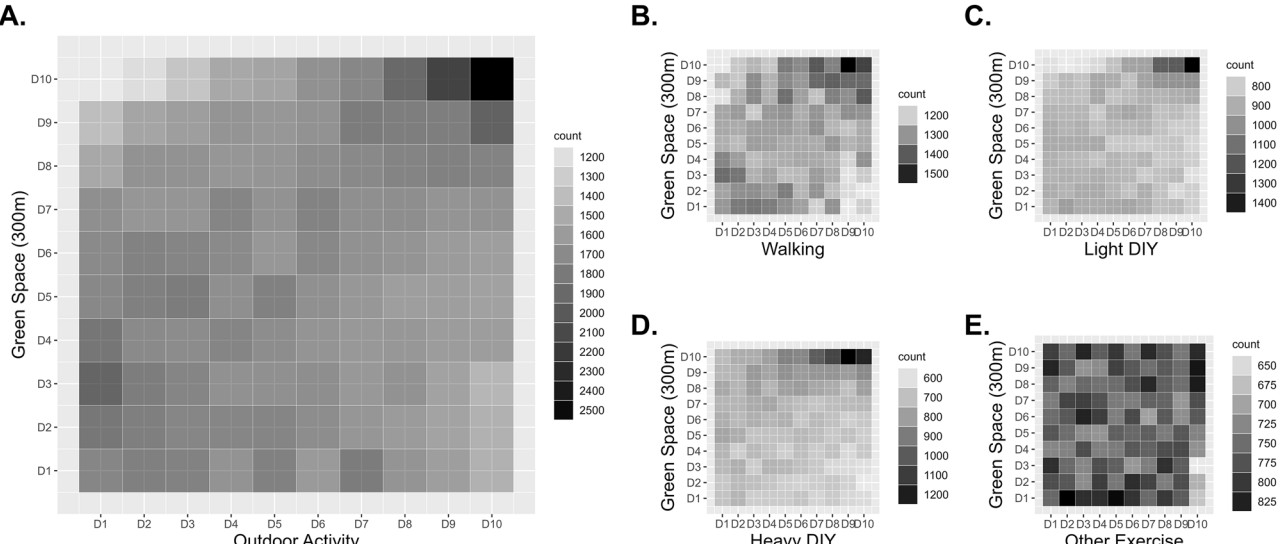

**Fig. 3 | Association between outdoor activities and percentage of residential green space at 300 m buffer zone. A** Association between the outdoor activity sum score and residential green space. **B** Association between walking and residential green space. **C** Association between light DIY activities and residential green space. **D** Association between heavy DIY activities and residential green space. **E** Association between other exercise and residential green space. *Notes.* This density plot illustrates the distribution of individuals based on their levels of outdoor activity and the percentage of green space within a 300 m buffer around their residence. Both the outdoor activity variable (measured in Metabolic Equivalent of Task [MET]-minutes) and the green space variable (measured as a percentage) were categorized into deciles, represented on the *x*-axis and *y*-axis, respectively. The plot displays density as the frequency of individuals within each grid cell, with darker colors indicating a higher concentration of individuals corresponding to specific deciles of outdoor activity and green space. From this plot, we can see that individuals living in areas with higher deciles of green space are more likely to engage in higher deciles of outdoor activity, such as walking, light DIY, and heavy DIY. Abbreviations. Light DIY light do-it-yourself activities, heavy DIY heavy do-it-yourself activities.

**Fig. 4 | Association between outdoor activities and all-cause dementia by residential green space at 300 m buffer zone.** Notes. This figure presents hazard ratios and its 95% confidence intervals of dementia outcome in relation to different types of outdoor activities. Numbers of included participants and cases, as well as exact hazard ratios and 95% confidence intervals, are provided in Supplementary Table 1. Each model was adjusted for individual-level factors (age, sex, education, smoking, cardiovascular health, somatic health, and residential length), index of multiple deprivations, air pollution, and noise level. More accessible green space was categorized as the top quartile of residential green space at 300 m buffer zone, and less accessible green space was identified with the remaining quartiles. Abbreviations. Light DIY light do-it-yourself activities, heavy DIY heavy do-it-yourself activities, HR Hazard Ratio, CI Confidence Interval.

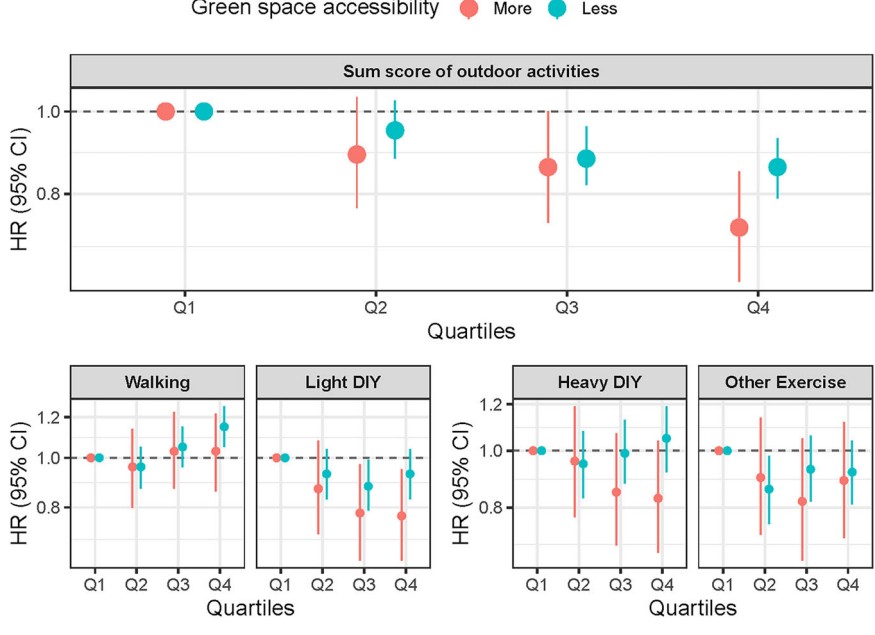

overall vascular health[33,34]. Our findings indicate that the association between increased engagement with outdoor activities and reduced risk of dementia may partially be explained by both neurodegenerative and vascular pathways. We observed a tendency for the combined effect of GS and outdoor activities on neurodegenerative markers, particularly hippocampal volume, but not on the WMH marker. The synergistic effects of outdoor activities and exposure to nature on brain health may be linked to increased neural activity in specific regions[35]. These underlying mechanisms need further investigation using a variety of biomarkers. Finally, future research is needed to better distinguish the types of activities associated with different GS areas. In this study, we assume that GS within a 300 m buffer is more relevant to daily routines and activities, such as walking or running, while a larger buffer (e.g., 1000 m) may reflect activities that occur on weekends or involve transportation modes like cycling. These assumptions warrant further investigation.

The strengths of the current study include its large sample size, a long follow-up period, comprehensive measurements of residential environments, incorporating assessments of GS at 300 m distance from the residential address. We included all-cause dementia as well as specific dementia subtypes and neuroimaging markers. In our analytical models,

**Table 2 | Association between outdoor activity, residential green space, and neuroimaging markers at the first neuroimaging visit**

| | Total sample (n = 36 854) | GS (300 m buffer) <53.28% (n = 27 641) | | GS (300 m buffer) ≥53.28% (n = 9213) | |
|---|---|---|---|---|---|
| | β coefficients (95% CI) | β coefficients (95% CI) | β coefficients (95% CI) | β coefficients (95% CI) | β coefficients (95% CI) |
| **Hippocampus** | Model 1[a] | Model 2[a] | Model 3[a] | Model 2[a] | Model 3[a] |
| Outdoor activity, categories | | | | | |
| First quartile | Ref. | Ref. | Ref. | Ref. | Ref. |
| Second quartile | 1.64 (−16.61, 19.88) | −4.93 (−25.62, 15.77) | −3.25 (−23.94, 17.44) | 15.44 (−23.28, 54.15) | 15.36 (−23.34, 54.06) |
| Third quartile | 23.48 (5.16, 41.81)* | 12.65 (−8.30, 33.60) | 15.18 (−5.75, 36.10) | 49.21 (11.06, 87.35)* | 48.83 (10.69, 86.96)* |
| Fourth quartile | 14.14 (−4.27, 32.46) | 6.32 (−14.95, 27.58) | 9.37 (−11.87, 30.60) | 29.64 (−7.74, 67.02) | 28.78 (−8.64, 66.20) |
| Outdoor activity, continuous[b] | 7.78 (0.57, 14.99)* | 3.88 (−4.42, 12.18) | 5.22 (−3.07, 13.50) | 15.22 (0.48, 29.95)* | 14.78 (0.02, 29.53)* |
| **Total gray matter** | Model 1[a] | Model 2[a] | Model 3[a] | Model 2[a] | Model 3[a] |
| Outdoor activity, categories | | | | | |
| First quartile | Ref. | Ref. | Ref. | Ref. | Ref. |
| Second quartile | 403.41 (−620.75, 1427.57) | 489.61 (−677.50, 1656.70) | 394.74 (−771.09, 1560.56) | 374.6 (−1775.24, 2524.52) | 355.1 (−1793.83, 2503.96) |
| Third quartile | 847.43 (−181.46, 1876.32) | 740.12 (−441.36, 1921.60) | 612.84 (−566.34, 1792.02) | 1357.0 (−761.09, 3475.46) | 1292.0 (−825.54, 3409.60) |
| Fourth quartile | 1614.48 (580.73, 2648.23)** | 1618.96 (419.96, 2817.95)* | 1431.69 (235.09, 2628.28)* | 1852.0 (−223.29, 3928.13) | 1832.0 (−245.31, 3910.22) |
| Outdoor activity, continuous[b] | 725.97 (321.19, 1130.74)*** | 696.76 (228.94, 1164.58)** | 619.59 (152.92, 1086.26)** | 900.7 (82.68, 1718.69)* | 889.4 (70.21, 1708.53)* |
| **WMH** | Model 1[a] | Model 2[a] | Model 3[a] | Model 2[a] | Model 3[a] |
| Outdoor activity, categories | | | | | |
| First quartile | Ref. | Ref. | Ref. | Ref. | Ref. |
| Second quartile | −0.02 (−0.05, 0.00) | −0.03 (−0.06, 0.01) | −0.04 (−0.06, −0.01)* | 0.01 (−0.04, 0.07) | 0.01 (−0.04, 0.06) |
| Third quartile | −0.05 (−0.07, −0.02)*** | −0.06 (−0.09, −0.03)*** | −0.06 (−0.09, −0.03)*** | 0.00 (−0.05, 0.05) | −0.00 (−0.05, 0.05) |
| Fourth quartile | −0.05 (−0.08, −0.03)*** | −0.06 (−0.09, −0.03)*** | −0.06 (−0.09, −0.03)*** | −0.02 (−0.07, 0.03) | −0.03 (−0.08, 0.02) |
| Outdoor activity, continuous[b] | −0.03 (−0.04, −0.02)*** | −0.03 (−0.04, −0.02)*** | −0.03 (−0.04, −0.02)*** | −0.01 (−0.03, 0.01) | −0.02 (−0.04, 0.00) |

[a]Model 1 was adjusted for individual-level factors (age at neuroimaging visit, sex, education, smoking, cardiovascular health, somatic health, residential length) and residential green space at 300 meter buffer zone.
[a]Model 2 was adjusted for individual-level factors and index of multiple deprivations.
[a]Model 3 was adjusted for individual-level factors, index of multiple deprivations, air pollution, and noise level.
[b]Continuous variable of outdoor activity was using log-transformed 100 MET-min/week. $*P < 0.05$; $**P < 0.01$; $***P < 0.001$
WMH white-matter hyperintensity, CI confidence interval.

we adjusted for both individual-level factors and residential-level variables to account for their influences. We further conducted sensitivity analyses to address the reverse causality between outdoor activity and dementia.

The current study has also several limitations. First, the study population consisted of volunteers, which may affect the generalizability of our findings[36]. Specifically, we observed a dementia incidence rate of approximately 3.0 per 1000 person-years in our population (aged ≥60 years), which is slightly lower than previously reported rates from UK primary care data for individuals aged 50 years and older between 2007 and 2015 (3.75 to 5.56 per 1000 person-years)[36]. This discrepancy may reflect selection bias, as UK Biobank participants tend to be healthier than population-based samples. Notably, in the UK Biobank data, men exhibited a higher risk of dementia than women, which contrasts with the typical pattern of higher dementia incidence observed among women. This reversal may be attributed to the greater burden of health-related conditions and less healthy lifestyle behaviours observed among men compared to women in the UK Biobank cohort[37]. Furthermore, beyond the healthy volunteer bias, large-scale volunteer databanks used in aging and age-related disease research often exhibit additional biases, such as overrepresentation of individuals with white ethnicity and higher education levels[38]. These biases may skew our understanding of aging by underrepresenting critical early and late life stages. Therefore, future research should interpret our findings with caution and aim to replicate them in diverse geographic and demographic populations to ensure broader generalizability. Second, we identified dementia outcomes using an algorithm that incorporated linked data from primary care records, hospital admissions (patient register), and the death register. Although previous studies have demonstrated that routinely collected healthcare datasets can accurately identify dementia cases, with positive predictive value estimates of up to 87%, limitations remain in the accurate classification of dementia subtypes[39]. This is partly due to the absence of specific ICD-10 codes for certain subtypes, such as dementia with Lewy bodies. Moreover, our findings related to frontotemporal dementia may be limited by statistical power, and caution is warranted when interpreting these non-significant results. Third, engagement in outdoor activities was assessed using self-reported data, which may introduce some level of reporting bias. Thus, future studies are warranted to develop specific measurements tailored to outdoor activities to capture more accurate and detailed information on subtypes of activities and exposure to GS. Fourth, when participants answered questions about engaging in other forms of exercise, their responses might include activities such as indoor swimming. Although this study performed specific analyses based on individual outdoor activities, future research is needed to quantify the relationship between outdoor and indoor activities and dementia. Fifth, we lack information regarding participants' movement patterns during the follow-up period and any changes in exposure resulting from relocation. Although we adjusted for residential length in our models, there may still be potential misclassification of exposure to GS and nature. In addition, the landscape data used in our study were derived from 2005, which may introduce measurement bias when analysing follow-up outcomes extending to 2022. Similarly, the noise and air pollution data were based on estimates from 2009 and 2005–2007, which may also contribute to measurement bias in our findings. Lastly, we did not adjust for BMI in our analysis, as it violated the proportional hazards assumption when included in the models. Notably, nearly 70% of the analytical sample was classified as overweight or obese, which may introduce selection bias. Future studies are needed to further explore the role of BMI in the relationship between outdoor activities and neurodegenerative diseases, such as incident dementia.

## Conclusions

In conclusion, engaging in outdoor activities may offer cognitive benefits by lowering the risk of dementia in older adults. Additionally, residing in areas with more accessible GS may further benefit the favourable effects on brain health by promoting outdoor activities. The synergistic effects of physical activities and exposure to nature may potentially help mitigate age-related neurodegeneration. Our study underscores the importance of integrating GS and natural environments into neighbourhood design. This suggests that urban planning and community development initiatives should prioritize the creation of accessible and well-designed green areas to promote brain health in the aging population.

## Data availability

We have provided derived variables for the administrators of the UK Biobank study. Researchers registered with UK Biobank can apply for access to the database by completing an application, which includes a summary of the research plan, data fields required, any new data or variables that will be generated, and payment to cover the incremental costs of servicing an application (https://www.ukbiobank.ac.uk/enable-your-research/applyfor-access). Source data underlying Figs. 2, 3, and 4 are provided in Supplementary Data 2, 3, and 4, respectively.

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

## Acknowledgements

This research has been conducted using data from UK Biobank, a major biomedical database. We would like to acknowledge all the researchers and participants involved in the UK Biobank. Additional acknowledgements go to the Swedish Research Council (VR), 2016-06658, 2022-01404, 2023-02601; the Swedish Research Council for Health, Working Life, and Welfare (FORTE): 2019-01120, 2020-00313; the Knowledge Foundation (KK-stiftelsen), 2018-0151, 2021-0002, 2022-0202, 2024-0066; Research Foundation Grant at Karolinska Institutet, 2022-02248; and Foundation for Geriatric Diseases at Karolinska Institutet, 2022-01286. MK was supported by Wellcome Trust, UK (221854/Z/20/Z), National Institute on Aging (NIH), US (R01AG056477) Medical Research Council, UK (MR/R024227/1, MR/Y014154/1), and Academy of Finland (350426).

## Author contributions

B.K., H.-X.W., Ö.E., J.W., H.W., M.K., and R.W. contributed to the study design, development of hypotheses, data interpretation, and critical review of the report. R.W. and B.K. were primarily responsible for drafting the manuscript. Data analyses were conducted by B.K. with support from R.W. B.K., R.W, and H.-X.W. had access to and verified the data. All authors (B.K., H.-X.W., Ö.E., J.W., H.W., M.K., and R.W.) reviewed the final version of the manuscript and approved the decision to submit it for publication.

## Funding

## Competing interests

The authors declare no competing interests.
