## [Transparent Peer Review file · Communications Medicine]

Outdoor Physical Activity, Residential Green Spaces and the Risk of Dementia in the UK Biobank Cohort

Corresponding Author: Dr Rui Wang

Version 0:

Reviewer comments:

Reviewer #1

(Remarks to the Author)

Please see my report attached

Reviewer #2

(Remarks to the Author)

Thank you for the opportunity to review this manuscript. This study aimed to evaluate the association between outdoor physical activities (OPA) and the risk of dementia in older adults using data from UK Biobank, an open access, long-term prospective study of > 500,000 adults aged 40-69 y at recruitment in 2006-10. Participants aged >60 y were included. Three objectives were examined using different analytical samples from the Biobank. Here are some comments which I hope will improve the manuscript. Of note, I can't evaluate the conversion method for OPA in MET and the interpretation of neuroimaging results given a lack of knowledge of these areas.

- The introduction is not thorough enough and seems intuitive. It would be relevant to report some evidence to further support the biological plausibility of this exposure on the risk of dementia, especially since this OPA has been little studied. What specifically are the positive effects on the brain observed in references 6 to 8? What are the long-term psychological effects of green exercise (GE)? Line 103-104: There is no reference to support the statement regarding the role of residential socioeconomic status or air pollution.

- Line 55 and line 114, and elsewhere if the case: The authors present the UK Biobank as a population-based cohort study, but this cohort study is not representative of the British population because it includes volunteers. Use prospective study instead.

- Line 121-122: Health related outcomes were tracked by linking them with national datasets. Please mention how valid these data sources are, especially for dementia onset. Death registers are usually not reliable for dementia diagnostic.

- The description of the flowchart in the text is confusing. It would be preferable to insert the flowchart in the manuscript and not in the appendix. Why does the analytical sample for objective 3 include people aged 40 and over? Why not consider those aged 60 and over like for Aims 1 and 2?

- Covariates: Why not have the BMI continuous in the models? Already the other variables are measured in a rather crude way (present vs. not present for cholesterol, diabetes, hypertension)? Don't you have biological data to support the presence of these co-morbidities in this databank?

- Statistical analysis: line 215: why not use age as time scale instead of calendar time? It's more appropriate for dementia.

- Have you considered doing mediation analyses to examine the link between biological mechanisms and your neurodegeneration biomarkers? This could be more sensitive than linear regression analyses.

- Only 3% of incident cases over 13 years in a sample of participants over 60 y, including more male than female. How does it compare to the general UK population? A selection bias is most probable. This needs to be discussed in detail.

- Line 316: Comparisons with other studies are too general (ref 10, 24 for example). Please properly present the results of other studies in the literature. As with the introduction, it's too vague.

- Line 381: It is good to recognize that the participation of volunteers is the first limitation of the UK Biobank. But not only does this affect the generalizability of the results, but also the internal validity of the study with the healthy volunteer bias (Brayne and Moffitt, 2022). This needs to be discussed.

- Table 1: Over 26% of participants have missing data for education. How were missing data treated in all regression

models?

Minor comments:

- Based on reference 18, it is reported that the cohort included adults from 37-73 y, but on the web site of the UK Biobank, it is reported 40-69 y. How can we explain this difference in age? What are the final figures?
- eTable 3: HR for AD for second quartile in less accessible green space: of 0.90 (0.96-1.28). The estimate is not included in the confidence interval.

Version 1:

Reviewer comments:

Reviewer #2

(Remarks to the Author)

I am very satisfied with the responses provided by the authors. Best of luck!

Referee expertise:

Referee #1: Neuropsychology; Dementia; cognitive decline

Referee #2: Epidemiology of Alzheimer's disease and cognitive decline; aging

Reviewers' comments:

Reviewer #1 (Remarks to the Author):

Re: COMMSMED-24-0971-T "Outdoor Physical Activity, Residential Green Spaces, and Risk of Dementia: The UK Biobank Observational Cohort Study"

The reported study addressed an important topic of research that is of significant global interest. The manuscript is generally well-written with good justifications for the chosen approaches. However, I did become confused in some places and felt that some clarification and further development of the writing (particularly in the discussion section) could increase the potential impact of this manuscript and make it more accessible to a broader readership. Modification of the statistical approach may be appropriate as well. Below I will detail my main comments and specific suggestions that may be helpful towards improving the manuscript.

Response: We appreciate the reviewer's positive feedback and agree that providing additional clarifications will further enhance the impact of our work. We will try our best to address all comments accordingly.

Main comments:

- Regarding 300 meters versus 1000 meters, I felt a bit confused by the approach taken and what exactly was meant by high accessibility. I wondered whether an opportunity to provide more specific insight with regard to the required proximity of green spaces was missed. It was indicated in multiple places that 1,000 meters provided similar results to 300 meters. Do the results indicate that there is no need to have green spaces within 300 meters as there is no greater advantage over 1000 meters? If there were no significant differences between the two, does it make sense to always present them separately, including in tables and figures. Including a large number of tables and figures unnecessarily risks losing readers. On the other hand, clarifying whether there is any advantage to having green spaces within 300 meters, as opposed to 1000 meters, may be useful information for city planners, but I don't think this was ever expressly addressed.

Response: We appreciate the reviewer's point and acknowledge that we did not clearly differentiate the similarities and differences between green space proportions in the 300-meter and 1000-meter buffers. Including both measures, along with a brief statement on their comparable findings, may have confused. The rationale for including both was to distinguish the types of activities associated with each area. For instance, the percentage of green space within a 300-meter buffer may be more relevant to daily activities or exercise routines, such as outdoor walking and running, while the 1000-meter buffer may capture activities occurring on weekends or involving biking and other forms of transportation daily. However, since this distinction is not the primary focus of our study and in line with the reviewer's suggestion, we have now focused primarily on presenting results for the 300-meter buffer, while the 1000-meter analysis has been moved to the supplementary materials and is briefly discussed.

See lines 463-468 in the tracking-change version. *“Finally, future research is needed to better distinguish the types of activities associated with different GS areas. In this study, we assume that GS within a 300-meter buffer is more relevant to daily routines and activities such as walking or running, while a larger buffer (e.g., 1000 meters) may reflect activities that occur on weekends or involve transportation modes like cycling. These assumptions warrant further investigation.”*

- I was not clear on the added value of splitting outdoor activity into quartiles. It did not seem to add anything beyond the results reported on lines 262 through 267.

Response: We understand the reviewer's concern, as the results for outdoor activity quartiles in Figure 1 (now updated as Figure 2) largely align with those observed when treating outdoor activity as a continuous variable. Yet the association between outdoor activity quartiles and vascular dementia displayed a slightly non-linear pattern. We have added explanations in lines 263-265 *“The sum of MET minutes of outdoor activities was modelled either as quartiles or as a log-transformed continuous variable to capture potential non-linear and linear associations with dementia outcomes”* and lines 323-326 *“Results indicated that higher quartiles of outdoor activity were associated with greater risk reductions for all-cause dementia, AD, and vascular dementia, with the strongest effects observed in the second quartile and smaller gains in the higher quartiles”*, to clarify the findings.

- In the discussion section, the focus on dementia risk seemed to be lost at points. Of note, lines 317 through 322 shift focus to dementia treatment followed by reference to mental well-being (line 324), which do not seem sufficiently relevant to me. My feeling was that this information could probably be deleted from the manuscript, else perhaps reworked and moved down to a later part of the discussion. In relation to this, take care to specify dementia risk where relevant (not just dementia), for example on line 326.

Response: Following the reviewer's suggestions, we have reorganized these sections. Please refer to lines line 395-415 in the tracking change version, *“Most previous studies focused on*

outdoor activity in people living with dementia. It has been hypothesized that engagement in activities outside of nursing homes might benefit patients with dementia in diverse ways.16 28 A systematic review focusing on intervention studies involving outdoor activities, e.g., aquatic exercise, wheelchair cycling, walking, and outdoor gardening, found preliminary evidence of benefits for individuals living with dementia in terms of psychological, physical, and physiological outcomes.28 However, the limited number of participants in the included studies for the joint analysis (n = 177) may restrict the generalizability of the findings.28 A narrative review focusing on long-term care facilities for individuals with dementia evaluated studies examining the impact of outdoor natural landscapes and activities, e.g., such as horticulture and gardening, on behavioural and psychological symptoms of dementia.16 The review summarized that outdoor natural landscape interventions—such as horticultural activities, therapeutic gardens, and green care farms—generally showed positive effects on agitation, apathy, and engagement among individuals with dementia.16 To our knowledge, the current study is the first of its kind to leverage a large population sample to explore the link between outdoor activities and long-term dementia risk.”

- I was surprised that there didn't seem to be any specific comments on whether the study was sufficiently powered to inform on frontotemporal dementia. With regard to the lower sample size, I wondered whether some of the wording might be a bit misleading, for example especially vascular dementia and AD among older adults. I had the feeling that the wording might be improved to better reflect the pattern of results, taking into consideration power. For example, on line 266 it might be useful to indicate the small sample size relative to the other dementia groups, to aid interpretation.

Response: We acknowledge the reviewer's observation. The results section has been rephrased. See lines 320-323, “The findings related to frontotemporal dementia may be due to insufficient statistical power.”, and discussed as limitation in lines 498-500 “Moreover, our findings related to frontotemporal dementia may be limited by statistical power, and caution is warranted when interpreting these non-significant results.”

- I struggled to understand what DIY has to do with green spaces. Is the idea that if you are generally more able-bodied in association with engaging in more outdoor exercise (due to the access to green spaces), you might also be more likely to engage in more DIY? There seemed to be an assumption that DIY takes place outdoors, whereas in my experience DIY often happens indoors. On a related note, I wondered whether the text description of the results presented in Figure 3, at the top of page 12, accurately capture the results presented in Figure 3. Further, the summary results at the top of Figure 3 seem somewhat surprising given the individual activity results presented below.

Response: We understood the reviewer's confusion about DIY. In the IPAQ questionnaires, light DIY and heavy DIY have been provided certain examples, we also stated in our text. See lines 181-184: “The evaluation of outdoor activities was based on the following items: (1) walking for pleasure, (2) participating in light DIY activities (do-it-yourself, e.g., home

improvement and gardening), (3) undertaking heavy DIY tasks (e.g., using heavy tools, weeding, lawn mowing, digging, and carpentry), and (4) participating in other forms of exercise (e.g., swimming and cycling).” We additionally acknowledged this in limitation, see lines 502-505: “Third, engagement of outdoor activities was assessed using self-reported data, which may introduce some level of reporting bias. Thus, future studies are warranted to develop specific measurements tailored to outdoor activities to capture more accurate and detailed information on subtypes of activities and exposure to GS.”

We followed the reviewer's suggestion and have rephrased and clarified the text. See lines 340-354 (Figure 3 has been updated as Figure 4), “Figure 4 presents the graphical results, while eTable 2 shows the detailed HRs and 95% CIs from the multivariable-adjusted model. Compared to the lowest quartile of total outdoor activities, the highest quartile was significantly associated with a reduced risk of dementia, with a greater magnitude of risk reduction observed in areas with more accessible green space (more accessible GS: HRQ4 vs. Q1 = 0.73, 95% CI = 0.63 — 0.85; less accessible GS: HRQ4 vs. Q1 = 0.86, 95% CI = 0.79 — 0.93). When examining individual outdoor activities, we observed that the associations of walking and light DIY with dementia risk varied slightly depending on the amount of residential GS. The association between higher quartiles of light DIY and reduced dementia risk appeared more pronounced among individuals living in areas with more accessible GS (more accessible GS: HRQ4 vs. Q1 = 0.77, 95% CI = 0.63 — 0.95; less accessible GS: HRQ4 vs. Q1 = 0.93, 95% CI = 0.83 — 1.04). Walking was not associated with dementia risk in areas with more accessible GS; however, compared to the lowest quartile, the highest quartile of walking was associated with an increased risk of dementia among those living in areas with less GS (HRQ4 vs. Q1 = 1.15, 95% CI = 1.05 — 1.26)”.

Specific suggestions/comments:

- Line 37 refers to 13-year risk but line 352 refers to 10 years.

Response: In the abstract, we described a 13-year period, while in the discussion, we referred to dementia risk occurring over 10 years later. To ensure consistency, we have changed “over 10 years” to “13 years”. See line 422.

- Do the acronyms OPA and GE refer to the same thing? If so, I recommend only using one of them to avoid reader confusion and reduce burden on readers (a larger number of acronyms puts extra burden on the reader, so good limit; regarding this, I wondered if it is necessary to define cSVD and I think its use on line 369 might confuse readers—you might want to add in parentheses evidenced by WMH).

Response: Green exercise (GE) and outdoor physical activity (OPA) are overlapping concepts with some distinctions. Since there is no objective or validated measure to quantify green exercise, which specifically refers to physical activity occurring in natural environments, we have used outdoor physical activity as a proxy. We understand the reviewer’s concern and

have only focused on outdoor activity through our text, removing the abbreviation to avoid confusion. Additionally, we have removed the abbreviation for cSVD and now refer only to WMH to improve clarity for readers.

- More volumes should perhaps read greater or higher volumes, for example on line 39 (more hippocampal...). Also, assuming I understood the design correctly, I believe reduced white matter hyperintensities should read fewer.

Response: We thank the reviewer, and the relevant text has been updated accordingly.

- Line 69. I was not clear what you meant by with high compared to low residential green space and wondered if it should instead read something like: living in areas with a high, compared low, percentage of residential green space?

Response: The reviewer is right, and we have updated the text according to the reviewer's suggestions.

- Line 75. Consider changing underscore urban planning to read underscore the importance of urban planning, or something along those lines.

Response: The text has been updated based on the reviewer's suggestion.

- I wondered whether you were justified in using the term well-designed (in reference to green spaces), given the design of your study.

Response: We acknowledge that the term 'well-designed' may require further clarification. The sentences have been rephrased to "These findings underscore the importance of urban planning that prioritizes accessible green spaces to promote brain health in the aging population."

- Line 105. Consider changing in neuroimaging studies to read in association with neuroimaging markers, or something along those lines.

Response: We thank the reviewer. The sentence has been rephrased as: "Moreover, few studies have examined the association between outdoor activities, residential green space, and neuroimaging outcomes".

- Take care not to reuse the names of figures and tables. For example, when referring to eTable 1, ensure that the e is always included to clearly differentiate it from Table 1, such as on line 146.

Response: All supplementary materials have been labelled in the text as eTable and eFigure following the reviewer's suggestion.

- Personally I struggled with the use of a space (1 000) rather than a comma (1,000) for numbers with more than three digits, especially when broken up at the end of line 120, but I appreciate that this is an accepted format in many contexts.

Response: Thank you for pointing this out. We have carefully reviewed the journal's formatting and updated all three-digit numbers to use commas as separators.

- At the bottom of Page 5, I wondered if it might help readers to insert within parenthetical statements aim 1, 2 and 3, to align with the description at the top of Page 5. Also I wondered if at baseline should be specified on line 128 after the word years.

Response: The text has been updated to: *"To (1) quantify the association between outdoor activities and dementia risk, and (2) examine whether GS modifies this association, we included 187,724 participants aged ≥ 60 years at baseline in the analytical sample. To (3) explore potential underlying mechanisms for this association, we included participants aged ≥ 40 years at baseline with available brain neuroimaging assessments in the analytical sample (n=36,854)."*

- Line 125. or missing value in the items of evaluating I presume should read: or missing values in the items evaluating

Response: The text has been updated.

- I wondered whether it is sufficiently useful to readers to provide Fields ID codes in the text, such as on line 149. I was not sure what to do with that information. I see on line 181 that they relate to the Biobank so perhaps the field IDs are required to enable replication, but in this case I think it would work best if you inform readers of their purpose, else it might be the case that you could move them to a supplement. Same comment regarding File ID on line 190.

Response: Following the reviewer's comments, all Field ID codes have been removed and organized into the supplementary materials (see eTables 1).

- I wondered whether 2005 might be considered a bit old with regard to land use data; if so, might be worth mentioning as a limitation. Also, regarding most of the participants being overweight, might be worth mentioning that as a limitation. Also using noise estimates from 2009 might be considered a limitation.

Response: we have addressed this in the limitations. See lines 513-521 *"In addition, the landscape data used in our study were derived from 2005, which may introduce measurement bias when analyzing follow-up outcomes extending to 2022. Similarly, the noise and air pollution data were based on estimates from 2009 and 2005-2007, which may also contribute to measurement bias in our findings. Lastly, we did not adjust for BMI in our analysis, as it violated the proportional hazards assumption when included in the models. Notably, nearly 70% of the analytical sample was classified as overweight or obese, which may introduce selection bias. Future studies are needed to further explore the role of BMI in the relationship between outdoor activities and neurodegenerative diseases, such as incident dementia."*

- Lines 196 through 197. I was not clear on whether that information was self-reported or obtained through medical records.

Response: We have updated the description. See lines 236-240: *“Based on multiple resources (e.g., blood markers in HbA1c, blood pressure, total cholesterol, use of medications, self-reported diagnosis), we additionally considered diabetes mellitus (present/not present), high cholesterol (present/not present), and hypertension (present/not present) to assess cardiovascular health.”*

- Delete the space before the closing parenthesis on lines 246 and 248.

Response: done.

- Line 294. I wondered whether you might want to clarify what exactly you mean by neurodegenerative markers.

Response: we have specified that the neurodegenerative marker is hippocampal volume.

- I wondered whether you might want to clarify what you mean by blue space for readers not familiar.

Response: We have added examples after the blue space, such as a lakeside, river or coast now.

- Line 310. Consider moving the words among older adults to line 309 after the word activities.

Response: done.

- On line 312, in reference to high accessibility, you might want to add in parentheses evidenced by... Readers might interpret high accessibility as referring to 300 meters versus 1000 meters.

Response: Done. Following the reviewer’s suggestion, the sentence has been updated to *“... in older adults residing in areas with high accessibility to GS at a 300-meter buffer distance”*.

- Line 329. Consider rewording a suggestive protective effect. Not sure if you might mean an apparent protective effect or perhaps evidence of a protective effect.

Response: We meant “an apparent protective effect”, which has been updated.

- Line 336. Again accessible neighborhood green space might be interpreted as referring to 300 as opposed to 1,000 meters. Also I wondered whether the wording is clear enough regarding nature components and other nature exposures, with respect to the results; it might be the case that readers would benefit from a bit more detail in the results section to ensure that they understand these results.

Response: Updated. See lines 424-427 *“The current study extends these findings by showing that among older adults aged 60 years and older, accessible neighborhood GS, as well as*

other nature components (e.g., rivers, lakes, oceans, farmlands, meadows), may magnify the impact of outdoor activities on dementia risk reduction".

- Line 338. I don't think the word increased is appropriate given the design. Perhaps greater?

Response: Done.

- Line 346. I wondered about the relevance of the word clean, and whether you might want to reframe that sentence to mesh better with your results, else perhaps you are wanting to join that sentence with the previous sentence. It feels like there is quite a bit of information before you get on to incorporating your results.

Response: We agree with the reviewer and the sentences have been rephrased.

- Line 357. I think the wording would benefit from clarification. I was not entirely clear on whether you were calling for a need to use objective measures or a need to develop them. Perhaps both?

Response: This has been updated by stating a need of objective measurements.

- Line 375. Consider clarifying the words at various distances to align more precisely with the methods.

Response: Have been clarified by showing 300-meter distance.

- Line 381 and 384. Consider inserting a references. Also, developing more tailored questionnaires seems like a separate issue to reporting bias.

Response: Following the reviewer's suggestion, we have added references and rephrase the sentences.

- Line 390. I found the word mobility confusing; you might want to clarify.

Response: Mobility has been explained as movement patterns.

- Line 396. Consider changing the second comma to the word by (to mesh better with the reported results).

Response: done.

- Table 1. I felt unclear on what Unknown means, for example not asked or data lost, and how that data might have affected the results. Seemed surprising to have such large numbers for education. I felt surprised by the consistency in the numbers for air pollution noise pollution and residential length, which made me wonder whether there might have been an error in entering the numbers; I was also surprised by the significant p-value for air pollution given that the mean values are identical.

Response: We thank the reviewer for noticing this. We mistakenly categorized individuals without a college or university degree as 'unknown.' We have now corrected this, updated the relevant numbers and results accordingly, and revised Table 1.

- Figures. I struggled with some of the small font sizes. It is important to enable as many readers as possible to be able to decipher the details. I also struggled a bit with the colored lines in eFigure 3; not sure if it might be helpful to use different colors, else perhaps increasing the line thickness would help.

Response: We thank the reviewer for pointing this out and agree that using a larger font would improve readability. The figures in the article have been updated accordingly.

- Figure 1: I did not find Q1 Q2 Q3 Q4 to be very mentally accessible and wondered whether a more impactful figure could be designed (not sure if it might be helpful to readers to graph the raw continuous data). I also felt confused as to why only two models were presented in this figure and why the details of model 2 matched model 3 elsewhere; please ensure clear.

Response: We have reorganized Figure 1 (now figure 2) and clarified the rationale for using quartiles (Q1–Q4) in the analysis in both statistical section and results *“The sum of MET minutes of outdoor activities was modelled either as quartiles or as a log-transformed continuous variable to capture potential non-linear and linear associations with dementia outcomes.”*. Furthermore, we have clarified that two models were used to examine the association between outdoor activities and dementia, while three models were applied in the stratified analysis to assess the modifying effects of green space on this association (see statistics, lines 265-268, 276-281).

- Figure 2. I was unable to understand these green space figures. Increasing the font sizes might help, but it may be that readers need a bit more guidance in interpreting these figures.

Response: We have increased the font sizes and expanded the notes to aid in the interpretation of the results.

Reviewer #2 (Remarks to the Author):

Thank you for the opportunity to review this manuscript. This study aimed to evaluate the association between outdoor physical activities (OPA) and the risk of dementia in older adults using data from UK Biobank, an open access, long-term prospective study of > 500,000 adults aged 40-69 y at recruitment in 2006-10. Participants aged >60 y were included. Three objectives were examined using different analytical samples from the Biobank. Here are some comments which I hope will improve the manuscript. Of note, I can't evaluate the

conversion method for OPA in MET and the interpretation of neuroimaging results given a lack of knowledge of these areas.

Response: We thank the reviewer for the positive feedback and constructive comments to improve our work.

- The introduction is not thorough enough and seems intuitive. It would be relevant to report some evidence to further support the biological plausibility of this exposure on the risk of dementia, especially since this OPA has been little studied. What specifically are the positive effects on the brain observed in references 6 to 8? What are the long-term psychological effects of green exercise (GE)? Line 103-104: There is no reference to support the statement regarding the role of residential socioeconomic status or air pollution.

Response: Thank you for the reviewer's suggestions. We have revised the introduction to provide a more focused and updated overview of the literature, including detailed findings on the mechanisms through which outdoor activity may benefit cognitive health. We also clarified the relevance and rationale behind our three study aims and reduced discussion of other environmental factors to maintain focus.

In summary, the introduction has been restructured to demonstrate the acute effects of outdoor activities on adult brain health, while also highlighting the need to investigate their long-term impact on dementia risk. We further underline the importance of exploring underlying mechanisms using neuroimaging markers and emphasize the need to assess whether residential green space modifies the relationship between outdoor activity and dementia or brain maintenance.

- Line 55 and line 114, and elsewhere if the case: The authors present the UK Biobank as a population-based cohort study, but this cohort study is not representative of the British population because it includes volunteers. Use prospective study instead.

Response: We agree, and we have replaced all instances of 'population-based cohort study' with 'prospective study' throughout the manuscript.

- Line 121-122: Health related outcomes were tracked by linking them with national datasets. Please mention how valid these data sources are, especially for dementia onset. Death registers are usually not reliable for dementia diagnostic.

Response: In our study, incident dementia was identified using an algorithm-based outcome, incorporating linked data from primary care records, hospital admissions (patient register), and the death register. Previous studies using UK Biobank data have demonstrated that the use of routinely collected healthcare datasets, both individually and in combination, can accurately identify dementia cases, with positive predictive value estimates ranging from

80% to 87% for all-cause dementia, vascular dementia, and Alzheimer's disease (10.1007/s10654-019-00499-1).

However, limitations remain. UK Biobank participants tend to be relatively young, and the dataset includes fewer cases of vascular dementia, mixed dementia, and dementia with Lewy bodies (DLB) than would be expected in a general older population. Additionally, the absence of a specific ICD-10 code for DLB means that such cases could only be identified through primary care data, making it more difficult to accurately classify dementia subtypes. We have expanded the discussion of these limitations regarding dementia outcomes in the manuscript. See lines 492-500 in the track-changing version, "*Second, we identified dementia outcomes using an algorithm that incorporated linked data from primary care records, hospital admissions (patient register), and the death register. Although previous studies have demonstrated that routinely collected healthcare datasets can accurately identify dementia cases, with positive predictive value estimates of up to 87%, limitations remain in the accurate classification of dementia subtypes. This is partly due to the absence of specific ICD-10 codes for certain subtypes, such as dementia with Lewy bodies. Moreover, our findings related to frontotemporal dementia may be limited by statistical power, and caution is warranted when interpreting these non-significant results*".

- The description of the flowchart in the text is confusing. It would be preferable to insert the flowchart in the manuscript and not in the appendix. Why does the analytical sample for objective 3 include people aged 40 and over? Why not consider those aged 60 and over like for Aims 1 and 2?

Response: Flowchart has now been introduced as Figure 1, and the description of the flowchart in the text has been updated.

We acknowledge that the original text was unclear and may have given the impression that we intended to include only participants aged 40 and over to address aim 3. In fact, our aim was to include all participants from the UK Biobank who underwent an MRI scan. We have now clarified this information in both the main text and the flowchart.

- Covariates: Why not have the BMI continuous in the models? Already the other variables are measured in a rather crude way (present vs. not present for cholesterol, diabetes, hypertension)? Don't you have biological data to support the presence of these co-morbidities in this databank?

Response: Including BMI as a continuous variable in the Cox regression model violated the proportional hazards assumption ($p < 0.05$). Therefore, BMI was adjusted in our models. This has been briefly mentioned in statistical method.

We apologize for any confusion our previous description may have caused. To clarify, the other variables were not based on self-reported data but were derived from multiple

sources, including biological measures (e.g., blood biomarkers), medication records, and clinical diagnoses. We have provided accurate information in the Methods section now, and even introduced Charlson comorbidity index in our updated analysis which involved in the current version. See lines 236-240, *“Based on multiple resources (e.g., blood markers in HbA1c, blood pressure, total cholesterol, use of medications, self-reported diagnosis), we additionally considered diabetes mellitus (present/not present), high cholesterol (present/not present), and hypertension (present/not present) to assess cardiovascular health. We used the Charlson Comorbidity Index (CCI), which summarizes 19 medical conditions based on ICD-10 codes from UK Biobank inpatient data, to assess participants’ somatic health.26”*

- Statistical analysis: line 215: why not use age as time scale instead of calendar time? It’s more appropriate for dementia.

Response: We acknowledge that using age as the time scale is often recommended, especially for studying age-related neurodegenerative diseases (Weuve J, et al, *Alzheimers Dement.* 2015). However, introducing age as the time scale also requires careful adjustment for generation/birth cohort effects, which are reflected by the calendar year in which participants entered/leave the cohort. For example, two individuals aged 70 could have joined the study up to 10 years apart, potentially introducing unmeasured cohort-related differences. The UK Biobank has a unique design, with a wide baseline age range (40 to 70+ years) and no sample-representative repeated follow-up visits during the study period. This makes it challenging to adequately adjust for birth cohort or generation effects when using age as the time scale. In this context, we chose to use calendar time as the time scale, which better preserves the chronological order of exposures and outcomes in real-world time. We adjusted for age at baseline, avoiding the complex and untestable assumption that risk at the same age is comparable across different birth cohorts. Additionally, we conducted sensitivity analyses by excluding participants who developed dementia or dropped out within the first 10 years of follow-up, to further minimize potential age-related bias and reverse causation. The results remained consistent, supporting the robustness of our main findings.

- Have you considered doing mediation analyses to examine the link between biological mechanisms and your neurodegeneration biomarkers? This could be more sensitive than linear regression analyses.

Response: We understand the reviewer’s concern regarding the mediation analysis. In this study, we explored potential biological mechanisms only after observing an association between outdoor activities and incident dementia. Specifically, we aimed to examine whether any neuroimaging markers might help explain this connection. Using linear regression analysis, we assessed the relationship between outdoor activity and neuroimaging markers, and then stratified the analysis by levels of green space. This approach was intended to test the modifying effect of green space, not a mediating effect. We did not

hypothesize that green space mediates the association between outdoor activity and neuroimaging markers, as we cannot determine the temporal sequence—whether living in greener areas promotes more outdoor activity or whether individuals who engage in outdoor activity are more likely to choose greener neighbourhoods.

- Only 3% of incident cases over 13 years in a sample of participants over 60 y, including more male than female. How does it compare to the general UK population? A selection bias is most probable. This needs to be discussed in detail.

Response: We thank the reviewer for this important suggestion and agree that it is essential to address the representativeness of our study population and the generalizability of our findings. We have specifically discussed this in our discussion, see lines xxx *“Specifically, we observed a dementia incidence rate of approximately 3 per 1000 person-years in our population (aged ≥ 60 years), which is slightly lower than previously reported rates from UK primary care data for individuals aged 50 years and older between 2007 and 2015 (3.75 to 5.56 per 1000 person-years).³⁶ This discrepancy may reflect selection bias, as UK Biobank participants tend to be healthier than population-based samples. Notably, in the UK Biobank data, men exhibited a higher risk of dementia than women, which contrasts with the typical pattern of higher dementia incidence observed among women. This reversal may be attributed to the greater burden of health-related conditions and less healthy lifestyle behaviours observed among men compared to women in the UK Biobank cohort.³⁷”*

- Line 316: Comparisons with other studies are too general (ref 10, 24 for example). Please properly present the results of other studies in the literature. As with the introduction, it's too vague.

Response: The comparisons have been expanded, and please see lines 394-415, *“Most previous studies focused on outdoor activity in people living with dementia. It has been hypothesized that engagement in activities outside of nursing homes might benefit patients with dementia in diverse ways.^{16 28} A systematic review focusing on intervention studies involving outdoor activities, e.g., aquatic exercise, wheelchair cycling, walking, and outdoor gardening, found evidence of benefits for individuals living with dementia in terms of psychological, physical, and physiological outcomes.²⁸ However, the limited number of participants in the included studies for the joint analysis (n = 177) ...”*

- Line 381: It is good to recognize that the participation of volunteers is the first limitation of the UK Biobank. But not only does this affect the generalizability of the results, but also the internal validity of the study with the healthy volunteer bias (Brayne and Moffitt, 2022). This needs to be discussed.

Response: We thank the reviewer for pointing this out. We have now added additional limitation based on the article by Brayne and Moffitt, see lines 486-492, *“Furthermore, beyond the healthy volunteer bias, large-scale volunteer databanks used in aging and age-related disease research often exhibit additional biases, such as overrepresentation of individuals with white ethnicity and higher education levels. These biases may skew our understanding of aging by underrepresenting critical early and late life stages.³⁸ Therefore, future research should interpret our findings with caution and aim to replicate them in diverse geographic and demographic populations to ensure broader generalizability.”*

- Table 1: Over 26% of participants have missing data for education. How were missing data treated in all regression models?

Response: We apologize for the earlier error regarding the missing data on education. After correcting the categorization, we now report 1.3% missing data for education. We have updated all analyses accordingly using the revised education variable. All missing values reported in Table 1 were treated as dummy categories in the regression models and are now appropriately addressed in the Methods section. See lines 280-281 *“In this study, missing data on confounding variables were addressed by including them as separate dummy categories in the regression models”*.

Minor comments:

- Based on reference 18, it is reported that the cohort included adults from 37-73 y, but on the web site of the UK Biobank, it is reported 40-69 y. How can we explain this difference in age? What are the final figures?

Response: We agree with the reviewer that this information may have been confusing to readers. In our analytical dataset, participants were aged 37 to 73 years at baseline, which is consistent with the information provided by the UK Biobank (see figure below). Only 7 individuals were younger than 40 at recruitment, and we excluded these individuals from our analyses when we define analytical sample. We also recognize the importance of ensuring consistency with previous UK Biobank publications, and have therefore revised the description in our manuscript accordingly (see line 146-149, *This study is based on the prospective longitudinal cohort—the UK Biobank. At baseline of the UK Biobank (2006—2010), over 500,000 participants aged ≥ 40 years attended to clinical examination in one of the 22 assessment centres throughout the UK.*) and updated the relevant figures throughout the text.

- eTable 3: HR for AD for second quartile in less accessible green space: of 0.90 (0.96-1.28).
 The estimate is not included in the confidence interval.

Response: Thank you to the reviewer for pointing this out. We have now updated the figures accordingly.

Re: COMMSMED-24-0971-T "Outdoor Physical Activity, Residential Green Spaces, and Risk of Dementia: The UK Biobank Observational Cohort Study"

The reported study addressed an important topic of research that is of significant global interest. The manuscript is generally well-written with good justifications for the chosen approaches. However, I did become confused in some places and felt that some clarification and further development of the writing (particularly in the discussion section) could increase the potential impact of this manuscript and make it more accessible to a broader readership. Modification of the statistical approach may be appropriate as well. Below I will detail my main comments and specific suggestions that may be helpful towards improving the manuscript.

Main comments:

- Regarding 300 meters versus 1000 meters, I felt a bit confused by the approach taken and what exactly was meant by *high accessibility*. I wondered whether an opportunity to provide more specific insight with regard to the required proximity of green spaces was missed. It was indicated in multiple places that 1,000 meters provided similar results to 300 meters. Do the results indicate that there is no need to have green spaces within 300 meters as there is no greater advantage over 1000 meters? If there were no significant differences between the two, does it make sense to always present them separately, including in tables and figures. Including a large number of tables and figures unnecessarily risks losing readers. On the other hand, clarifying whether there is any advantage to having green spaces within 300 meters, as opposed to 1000 meters, may be useful information for city planners, but I don't think this was ever expressly addressed.
- I was not clear on the added value of splitting outdoor activity into quartiles. It did not seem to add anything beyond the results reported on lines 262 through 267.
- In the discussion section, the focus on dementia risk seemed to be lost at points. Of note, lines 317 through 322 shift focus to dementia treatment followed by reference to mental well-being (line 324), which do not seem sufficiently relevant to me. My feeling was that this information could probably be deleted from the manuscript, else perhaps reworked and moved down to a later part of the discussion. In relation to this, take care to specify dementia *risk* where relevant (not just dementia), for example on line 326.
- I was surprised that there didn't seem to be any specific comments on whether the study was sufficiently powered to inform on frontotemporal dementia. With regard to the lower sample size, I wondered whether some of the wording might be a bit misleading, for example *especially* vascular dementia and AD among older adults. I had the feeling that the wording might be improved to better reflect the pattern of results, taking into consideration power. For example, on line 266 it might be useful to indicate the small sample size relative to the other dementia groups, to aid interpretation.
- I struggled to understand what DIY has to do with green spaces. Is the idea that if you are generally more able-bodied in association with engaging in more outdoor exercise (due to the access to green spaces), you might also be more likely to engage in more DIY? There seemed to be an assumption that DIY takes place outdoors, whereas in my experience DIY often happens indoors. On a related note, I wondered whether the text

description of the results presented in Figure 3, at the top of page 12, accurately capture the results presented in Figure 3. Further, the summary results at the top of Figure 3 seem somewhat surprising given the individual activity results presented below.

Specific suggestions/comments:

- Line 37 refers to 13-year risk but line 352 refers to 10 years.
- Do the acronyms OPA and GE refer to the same thing? If so, I recommend only using one of them to avoid reader confusion and reduce burden on readers (a larger number of acronyms puts extra burden on the reader, so good limit; regarding this, I wondered if it is necessary to define cSVD and I think its use on line 369 might confuse readers—you might want to add in parentheses evidenced by WMH).
- *More* volumes should perhaps read *greater* or *higher* volumes, for example on line 39 (more hippocampal...). Also, assuming I understood the design correctly, I believe *reduced* white matter hyperintensities should read *fewer*.
- Line 69. I was not clear what you meant by *with high compared to low residential green space* and wondered if it should instead read something like: living in areas with a high, compared low, percentage of residential green space?
- Line 75. Consider changing *underscore urban planning* to read underscore the importance of urban planning, or something along those lines.
- I wondered whether you were justified in using the term *well-designed* (in reference to green spaces), given the design of your study.
- Line 105. Consider changing *in neuroimaging studies* to read in association with neuroimaging markers, or something along those lines.
- Take care not to reuse the names of figures and tables. For example, when referring to eTable 1, ensure that the e is always included to clearly differentiate it from Table 1, such as on line 146.
- Personally I struggled with the use of a space (1 000) rather than a comma (1,000) for numbers with more than three digits, especially when broken up at the end of line 120, but I appreciate that this is an accepted format in many contexts.
- At the bottom of Page 5, I wondered if it might help readers to insert within parenthetical statements aim 1, 2 and 3, to align with the description at the top of Page 5. Also I wondered if at baseline should be specified on line 128 after the word years.
- Line 125. *or missing value in the items of evaluating* I presume should read: or missing values in the items evaluating
- I wondered whether it is sufficiently useful to readers to provide *Fields ID* codes in the text, such as on line 149. I was not sure what to do with that information. I see on line 181 that they relate to the Biobank so perhaps the field IDs are required to enable replication, but in this case I think it would work best if you inform readers of their purpose, else it might be the case that you could move them to a supplement. Same comment regarding *File ID* on line 190.
- I wondered whether 2005 might be considered a bit old with regard to land use data; if so, might be worth mentioning as a limitation. Also, regarding most of the participants

being overweight, might be worth mentioning that as a limitation. Also using noise estimates from 2009 might be considered a limitation.

- Lines 196 through 197. I was not clear on whether that information was self-reported or obtained through medical records.
- Delete the space before the closing parenthesis on lines 246 and 248.
- Line 294. I wondered whether you might want to clarify what exactly you mean by neurodegenerative markers.
- I wondered whether you might want to clarify what you mean by blue space for readers not familiar.
- Line 310. Consider moving the words *among older adults* to line 309 after the word activities.
- On line 312, in reference to high accessibility, you might want to add in parentheses evidenced by... Readers might interpret high accessibility as referring to 300 meters versus 1000 meters.
- Line 329. Consider rewording *a suggestive protective effect*. Not sure if you might mean an apparent protective effect or perhaps evidence of a protective effect.
- Line 336. Again *accessible* neighborhood green space might be interpreted as referring to 300 as opposed to 1,000 meters. Also I wondered whether the wording is clear enough regarding *nature components* and *other nature exposures*, with respect to the results; it might be the case that readers would benefit from a bit more detail in the results section to ensure that they understand these results.
- Line 338. I don't think the word increased is appropriate given the design. Perhaps greater?
- Line 346. I wondered about the relevance of the word clean, and whether you might want to reframe that sentence to mesh better with your results, else perhaps you are wanting to join that sentence with the previous sentence. It feels like there is quite a bit of information before you get on to incorporating your results.
- Line 357. I think the wording would benefit from clarification. I was not entirely clear on whether you were calling for a need to use objective measures or a need to develop them. Perhaps both?
- Line 375. Consider clarifying the words at various distances to align more precisely with the methods.
- Line 381 and 384. Consider inserting a references. Also, developing more tailored questionnaires seems like a separate issue to reporting bias.
- Line 390. I found the word mobility confusing; you might want to clarify.
- Line 396. Consider changing the second comma to the word by (to mesh better with the reported results).
- Table 1. I felt unclear on what *Unknown* means, for example not asked or data lost, and how that data might have affected the results. Seemed surprising to have such large numbers for education. I felt surprised by the consistency in the numbers for air pollution noise pollution and residential length, which made me wonder whether there might have been an error in entering the numbers; I was also surprised by the significant p-value for air pollution given that the mean values are identical.

- Figures. I struggled with some of the small font sizes. It is important to enable as many readers as possible to be able to decipher the details. I also struggled a bit with the colored lines in eFigure 3; not sure if it might be helpful to use different colors, else perhaps increasing the line thickness would help.
- Figure 1: I did not find Q1 Q2 Q3 Q4 to be very mentally accessible and wondered whether a more impactful figure could be designed (not sure if it might be helpful to readers to graph the raw continuous data). I also felt confused as to why only two models were presented in this figure and why the details of model 2 matched model 3 elsewhere; please ensure clear.
- Figure 2. I was unable to understand these green space figures. Increasing the font sizes might help, but it may be that readers need a bit more guidance in interpreting these figures.